# OPEN TEACH: A Versatile Teleoperation System for Robotic Manipulation

**Aadhithya Iyer**[*]
New York University

**Zhuoran Peng**
New York University

**Yinlong Dai**
New York University

**Irmak Guzey**
New York University

**Siddhant Haldar**
New York University

**Soumith Chintala**
Meta AI

**Lerrel Pinto**
New York University

https://open-teach.github.io/

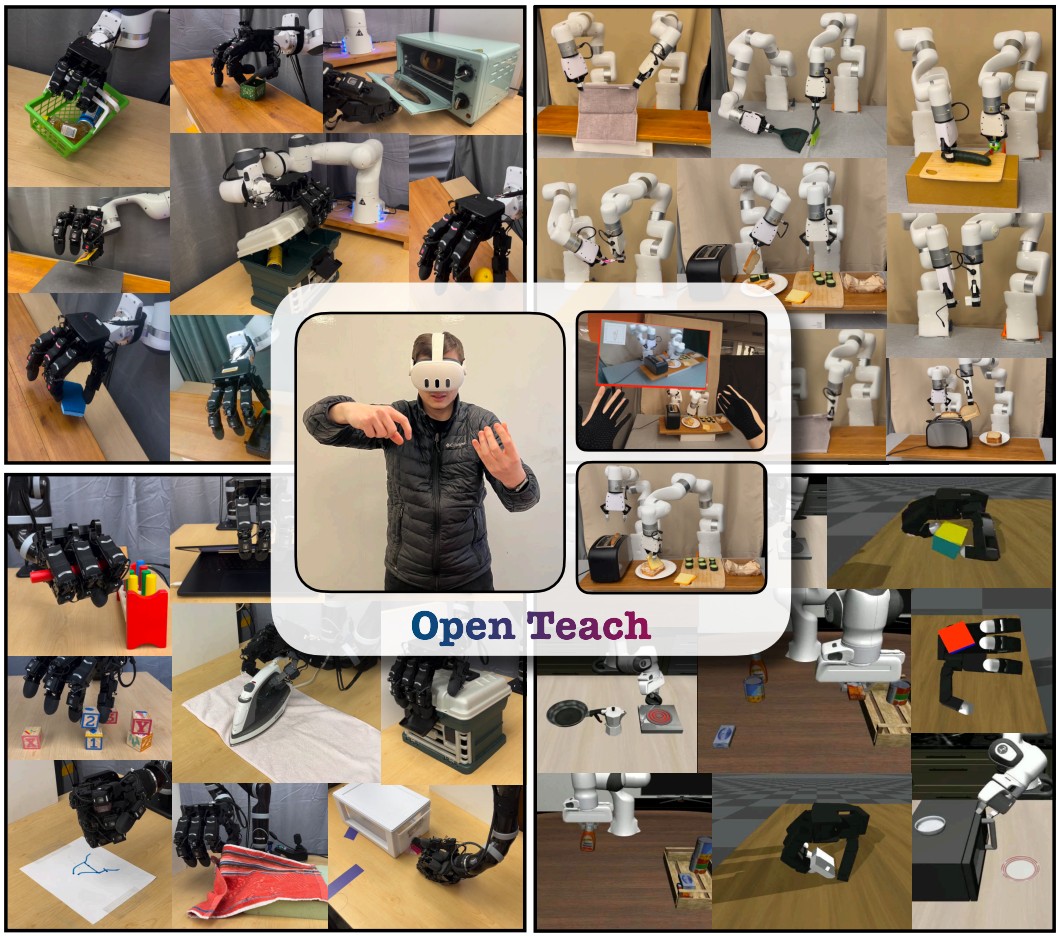

Figure 1: We present OPEN TEACH, a unified robot teleoperation framework that supports multiple arms and hands, allows mobile manipulation, is calibration-free, and works across both simulation and real-world environments. OPEN TEACH uses a VR headset for teleoperation, offers low latency and high-frequency visual feedback. This high-frequency operation allows human users to correct for robot errors in real time, facilitating the execution of intricate and long-horizon tasks. From *making a sandwich* and *ironing cloth* to *placing items in a basket and lifting it* and *approaching a cabinet and opening it*, OPEN TEACH delivers a comprehensive, user-friendly teleoperation experience for a wide range of applications. OPEN TEACH is fully open-source.

---

[*]Correspondence to: aadhithya.iyer@nyu.edu

8th Conference on Robot Learning (CoRL 2024), Munich, Germany.

**Abstract:** Open-sourced, user-friendly tools form the bedrock of scientific advancement across disciplines. The widespread adoption of data-driven learning has led to remarkable progress in multi-fingered dexterity, bimanual manipulation, and applications ranging from logistics to home robotics. However, existing data collection platforms are often proprietary, costly, or tailored to specific robotic morphologies. We present OPEN TEACH, a new teleoperation system leveraging VR headsets to immerse users in mixed reality for intuitive robot control. Built on the affordable Meta Quest 3, which costs $500, OPEN TEACH enables real-time control of various robots, including multi-fingered hands, bimanual arms, and mobile manipulators, through an easy-to-use app. Using natural hand gestures and movements, users can manipulate robots at up to 90Hz with smooth visual feedback and interface widgets offering closeup environment views. We demonstrate the versatility of OPEN TEACH across 38 tasks on different robots. A comprehensive user study indicates significant improvement in teleoperation capability over the AnyTeleop framework. Further experiments exhibit that the collected data is compatible with policy learning on 10 dexterous and contact-rich manipulation tasks. Currently supporting Franka, xArm, Jaco, Allegro, and Hello Stretch platforms, OPEN TEACH is fully open-sourced to promote broader adoption. Videos are available at https://open-teach.github.io/.

**Keywords:** Low-Cost Teleoperation, Imitation Learning, Robot Manipulation

## 1 Introduction

The integration of learning-based methods has sparked a revolution in robotics, significantly enhancing capabilities in manipulation [1, 2, 3, 4], locomotion [5, 6, 7, 8], and aerial robotics [9, 10, 11]. More recent work has been making advancements in complex single-task behavior learning [12, 13, 2], multitask scenarios [14, 15], multimodal behavior learning [16, 17, 18, 19], and efficient fine-tuning of pretrained behavior models [20, 21, 22]. A fundamental requirement across all these threads of research is the need to collect data in the form of task demonstrations.

Commonly used teleoperation systems include devices such as joysticks and 3D spacemouses [23, 24], commercial VR headsets [25, 26, 27, 13, 28, 29], kinesthetic teaching [30], and phone teleoperation [31]. The aforementioned devices are cost-effective and easy to set up. However, they are often unintuitive to use and require extensive user-training to demonstrate intricate motions. Recently proposed exoskeleton-based teleoperation frameworks like ALOHA [2], GELLO [32], and AirExo [33] attempt to alleviate this problem by having the human teleoperator directly control a kinematically isomorphic version of the same robot arm. These frameworks directly impose the kinematic constraints of the robot arm during teleoperation making it more compatible and intuitive to control the motion of the robot. Although highly effective, these systems can require an additional robot for each robot being controlled, have high initial setup costs, and are designed for specific robot morphologies.

The challenge of easy-to-use teleoperation devices is more apparent in dexterous manipulation problems [34, 35, 27, 13], owing to the high dimensional action space. Such frameworks typically involve the use of expensive gloves [36, 37, 38], extensive calibration processes [34, 27], or are susceptible to monocular occlusions [27].

In this work, we present OPEN TEACH, an open-source framework for robot teleoperation that supports a variety of robots, including bimanual and multi-finger manipulation, all at a price of $500. As shown in Figure 1, OPEN TEACH uses a VR headset (e.g. Quest 3) to put users / teachers in an immersive virtual world where they can view a robotic scene both through their eyes, via visual passthrough, as well as realtime streams from the robot's cameras. To control the robot, users can simply use hand gestures, which are detected using onboard hand-pose estimators at 90Hz. As a result, even though OPEN TEACH is kinematics-unaware, the high frequency execution and improved hand pose detection accuracy enables users to collect high-quality real-time robot demonstrations.

We experimentally evaluate OPEN TEACH on 38 tasks across single arm, bimanual, multi-fingered, and mobile manipulation robot setups in both simulation and the real world. The tasks span from tabletop manipulation to contact-rich dexterous manipulation. Across different robot morphologies, we find that users can provide demonstrations at speeds on par with robot-specific teleoperation systems and significantly faster than general-purpose systems like AnyTeleop [35]. Importantly, policies trained on the data collected achieve an average success rate of 86% on 10 tasks in simulation and the real world, validating the utility of policy learning using OPEN TEACH. The contributions of this work is summarized as follows:

1. We present OPEN TEACH, an open-source system for plug-and-play teleoperation framework suitable for collecting demonstrations across different robot morphologies in both simulation and the real world.

2. We experimentally show that the demonstrations collected by OPEN TEACH can be used to train effective, general-purpose manipulation behaviors.

3. Our user study on 15 users highlights the efficacy of OPEN TEACH for both experienced and new users.

OPEN TEACH is fully open-source with the mixed reality API, policy training code, and demonstrations collected using OPEN TEACH available at `https://open-teach.github.io/`.

## 2 Related Work

### 2.1 Robot-Specific Teleoperation

Teleoperation, as a medium for human-robot interaction, has been a crucial part of robotics. Recent strides in learning-based methods demand extensive data collection, giving rise to diverse teleoperation systems — joysticks and spacemouses [23, 24], VR controllers [25, 26, 27, 13, 28, 29], RGB cameras [34, 39, 35, 40], IMU sensors [41, 42, 43], kinesthetic teaching [30], phone teleoperation [31], gloves [36, 37, 38], marker-based motion capture systems [44, 45], and reacher-grabber sticks [46, 4]. However, several challenges remain in the robot-specific nature of these devices. For instance, devices like joysticks, spacemouses, and phones are limited to controlling robot arms due to their lack of fidelity for multi-fingered hands. There have been systems developed to map the human hand pose to the robot pose [27, 13, 35, 47, 48, 49], but they are often restricted to only controlling robot hands. Further, all of these frameworks lack awareness of the robot's kinematic constraints, leading to challenges in intuitive control, especially in complex poses. As a solution, there are more conventional but expensive exo-skeleton based teleoperation systems [50, 51, 52] that use a second arm for controlling the manipulator arm. Recently proposed ALOHA [2, 53] affirms the effectiveness of this approach through impressive results in fine-grained bimanual manipulation. However, these systems require an exact copy of the manipulator robot arm, rendering them costly and less practical for heavier robots. In addressing these issues, GELLO [32] and AirExo [33] introduce exo-skeleton teleoperation frameworks, utilizing a kinematically isomorphic variant of the robot arm. This approach proves more affordable and lightweight, enhancing usability for humans. Despite their success in fine-grained manipulation tasks, these solutions are constrained to robot arms and face challenges in extending seamlessly to control robot hands. For multi-fingered robot hands, gloves, vision-based, and VR-based methods have been employed. These systems either assume a fixed robot arm [27, 13] or are tied to a specific robot setup [34, 54], making them difficult to transfer to new arm-hand systems and new environments.

### 2.2 Unified Teleoperation Frameworks

The robotics community has often sought to develop versatile systems that operate across diverse environments and robots [55, 14, 56]. Leveraging the success of learning-based methods, achieving this requires teleoperation systems adaptable to various robot variants, allowing for abundant data collection with minimal setup costs. While methods combining robot arms with multi-fingered hands exist [34, 39, 28, 57, 54], their applicability across robot variants remains unclear.

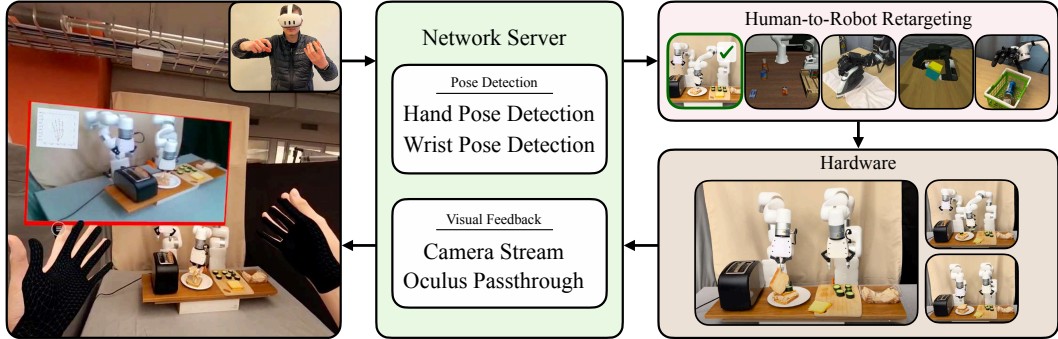

Figure 2: Overview of the teleoperation module in OPEN TEACH. Provided a hand and wrist pose within the VR interface, the controller transmits keypoint data to the robot's server. The server then transforms and retargets these key points to align with the specific robot setup. Real-time visual feedback of the teleoperated scene is promptly relayed back to the VR headset.

AnyTeleop [35] makes progress in this direction by proposing a robot-agnostic system compatible with multiple hands and arms. We build upon this idea in creating OPEN TEACH.

## 3   OPEN TEACH

In OPEN TEACH, a user wears a Virtual Reality (VR) headset to provide demonstrations to a robot. This involves creating a virtual world for teaching, retargeting the teacher's hand and wrist pose to the robot joints, and finally controlling the robot. We compare OPEN TEACH with various other teleoperation systems across a variety of robot types (Appendix **??**) and observe that OPEN TEACH is the only framework that enables controlling multiple arms, hands, and mobile manipulators, is calibration-free, and is completely open-source. In this section, we provide details about the VR-based teleoperation setup and the system design that enables data collection using this framework.

### 3.1   Placing an Operator in a Virtual World

We use the Meta Quest 3 VR headset to place the human teacher in a virtual world. The headset surrounds the human in a virtual environment at a resolution of $2064 \times 2208$ and a native refresh rate of 90Hz. The base version of this headset is affordable at $499 and is relatively light at 513g. Compared to the Meta Quest 2 VR headset used in prior work [13], the Quest 3 provides a full-color passthrough allowing the human to get a direct view of the robot setup during teleoperation. These features, especially the full-color passthrough, allow for a comfortable and intuitive operation by the user. Additionally, similar to Arunachalam et al. [13], the Quest 3 API interface allows for creating custom mixed reality worlds that visualize the robotic system along with diagnostic panels in VR. Examples of virtual scenes have been shown in Fig. 2 and Fig. 3. It is important to highlight the exceptional clarity of the scene passthrough visible in Quest 3.

### 3.2   Pose Estimation with VR Headsets

Similar to  Arunachalam et al. [13], we directly use the in-built hand pose estimator [58] of the Quest 3 using 2 monochrome cameras. This is significantly more robust compared to single camera alternatives [59]. Further, since the cameras are internally calibrated, they do not require specialized calibration routines that are needed in prior multi-camera teleoperation frameworks [34, 35]. Also, since the hand-pose estimator is integrated into the device, it can stream real-time hand poses at 90Hz. This alleviates the challenge of obtaining hand poses at both high accuracy and high frequency, as reported in prior work [34, 27].

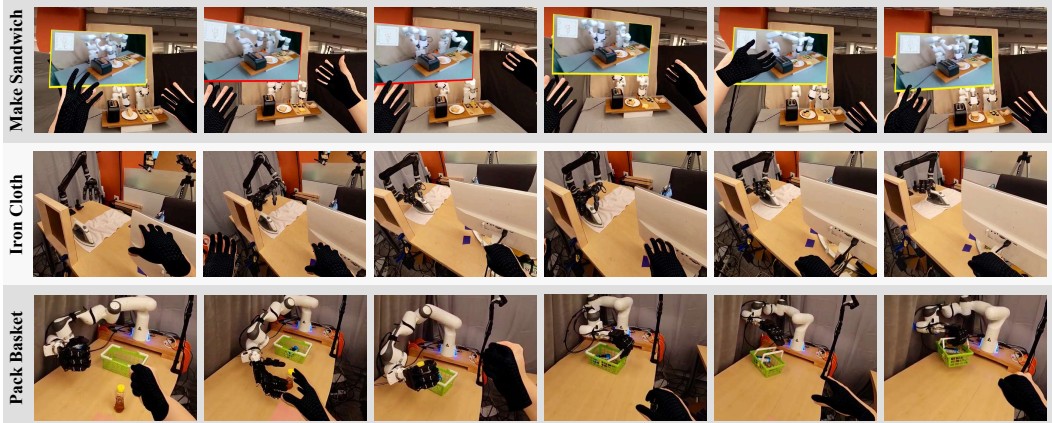

Figure 3: The demonstration collection process as viewed from within the VR application. Shown here is one task being performed for each real-world setup. High resolution images streamed at 90 Hz to the VR application allow for an immersive experience and enable reactive control by the user.

### 3.3 Human to Robot Pose Retargeting

The inbuilt hand pose estimate from the VR headset provides us with the joint positions of all the fingers of the human hand and the wrist. With this information, we can design wrappers that use combinations of these joint positions to map the human hand poses to the robot poses for any given robot morphology. In this work, we use a variety of robot arms, each with either a 2-fingered gripper or a multi-fingered robot hand. Below, we provide the wrapper design for each robot morphology used in this work.

Robot Arm We establish a 3D coordinate system using the wrist keypoint and knuckle points of the index and pinky fingers to define a 2D plane along the palm and a perpendicular third axis. The wrist position maps to the robot end effector position. Changes in the orientation of this hand coordinate system over time map to adjustments in end effector orientation.

Robot Hand We use the teacher's hand pose obtained from the VR to compute the individual joint angles in the teacher's hand. Given these joint angles, a straightforward method of retargeting is to directly command the robot's joints to the corresponding angles. In practice, this works well for all fingers except the thumb. To address this, we improve upon Arunachalam et al. [13], where the spatial coordinate of the teacher's thumb tip is mapped to that of the robot hand and then an inverse kinematics solver is used to compute the joint angles of the thumb. More details about the improvement in thumb retargeting have been included in Appendix 5.1. Since the Allegro hand does not have a pinky finger, we ignore the teacher's pinky joints.

Two-fingered gripper To detect the opening and closing of the two-fingered gripper, we utilize the pinch between the pinky finger and the thumb. The pinch is detected by computing the distance between the tips of the two fingers and setting a threshold on the pinch distance. We use a toggle mechanism for opening and closing the gripper where each pinch indicates toggling to the alternate state of the gripper.

Mobile manipulator The same 3D coordinate system established for controlling robot arms is used for mapping the wrist's movements to actions of the mobile robot. When the wrist moves forward, it extends the robot's arm, enabling it to reach farther. Vertical wrist movements adjust the robot's height, while lateral wrist movements cause the robot to move sideways by controlling its wheels. The 3D transformations across time are mapped to end effector orientation changes. The opening and closing of the gripper are controlled through the pinch between the index finger and the thumb.

These are just a few examples of how the hand pose data can be used to obtain a mapping between the human hand and a robot. The simplicity of the proposed framework has been summarized in Code Snippet **??** in the appendix. The primary idea behind OPEN TEACH is that given a VR headset, the end-user has the flexibility to design their own human-to-robot wrappers using the

human hand poses as input. The framework has been designed for simple integration with any robot setup, allowing robot teleoperation with real-time streaming (up to 90Hz) and low-latency visual feedback. This significantly reduces the initial setup cost as compared to prior exoskeleton-based teleoperation frameworks like GELLO [32] and AirExo [33].

## 3.4 Robot Control

Achieving minimal error and low latency is pivotal for OPEN TEACH to facilitate the intuitive teleoperation of the robot hand by the human teacher. In this study, we employ the Allegro Hand as our robotic hand, controlling it asynchronously through the ROS [60] communication framework. Using the computed robot joint positions from the retargeting procedure, a PD controller outputs desired torques at a frequency of 300Hz. To mitigate steady-state error, we include a gravity compensation module to compute offset torques.

trim

We use three different robot arms for our evaluations — xArm, Franka Emika Panda, and Kinova Jaco. We use different controllers for each. The xArm is directly controlled through the official xArm API [61]. For the Franka Emika Panda, we use the Deoxys controller [62]. For the Kinova Jaco, we use the controller open-sourced by Arunachalam et al. [27]. The streaming frequencies for all robots is provided in Table 1. Such high frequency teleoperation allows the human teacher to see the robot move in real time and immediately correct execution errors in the robot. The Hello Stretch is controlled at 5Hz using the controller released by Shafiullah et al. [4]. Further, we acknowledge the fact that the human hand possesses fewer degrees of freedom than the robot. In response, we introduce a pause functionality, allowing the teacher to momentarily halt teleoperation, reorient themselves, and resume the process. We also implement a resolution adjustment feature, which provides a performance boost for high-precision tasks such as delicately picking up a tea sachet. Details about these implementations have been included in Appendix 5.1.

## 4 Experiments

Our experiments and tasks are designed to answer the following questions: (1) How versatile is OPEN TEACH across a range of robotics setups? (2) How successful are policies trained with OPEN TEACH? (3) Can OPEN TEACH be used for performing complex, long-horizon tasks? (4) How intuitive is the system for new users?

### 4.1 Experimental Setup

We evaluate the versatility of OPEN TEACH by using it to collect demonstrations on six different setups — four in the real world and two in simulation. Each setup is a combination of a variant of a robot arm with either an Allegro Hand or a 2-fingered gripper. The real-world setups include:

1. **Franka-Allegro:** A Franka Arm with an Allegro Hand having the Xela tactile sensors.
2. **Kinova-Allegro:** A Kinova Jaco Arm with an Allegro Hand with the Xela tactile sensors.
3. **Bimanual:** 2 xArm7 robot arms with 2-fingered grippers.
4. **Stretch:** Hello Stretch mobile manipulator with a 2-fingered gripper.

The Franka-Allegro and Kinova-Allegro comprise a single Intel Realsense camera for data collection, whereas the Bimanual setup collects data from 5 different cameras. The Stretch has an iPhone attached to the wrist for data collection [4]. The simulated environments include:

1. **Allegro Sim:** A floating Allegro Hand capable of performing static and dynamic tasks.
2. **LIBERO Sim [23]:** A Franka Arm with a 2-fingered gripper placed in varied scenes.

We demonstrate the usefulness of the collected data by training visual and visuotactile policies using behavior cloning [63] and inverse RL [64, 65].

Table 1: Teleoperation Frequency across all robots.

| Domain | Robot Setup | Stream Frequency (in Hz) | |
| --- | --- | --- | --- |
| | | **Arm** | **End Effector** |
| Real | Franka-Allegro | 60 | 60 |
| | Kinova-Allegro | 60 | 60 |
| | Bimanual | 90 | 90 |
| | Stretch | 5 | 5 |
| Sim | Allegro Sim | 60 | 60 |
| | LIBERO Sim | 20 | 20 |

Table 2: Performance of policies learned on data collected through OPEN TEACH. For Franka-Allegro, Allegro Sim, and Libero Sim, TAVI [66], FISH [21] and BC were respectively used to train policies.

| Robot Setup | Task | Number of Demos | Success Rate |
| --- | --- | --- | --- |
| Franka-Allegro | Open Box | 3 | 9/10 |
| | Grasp Sponge | 6 | 7/10 |
| | Pick Up Tea Sachet | 4 | 7/10 |
| | Grasp Object and Twist | 6 | 8/10 |
| Allegro Sim | Flip Cube | 6 | 10/10 |
| | Flip Sponge | 6 | 10/10 |
| | Pinch Grasp | 6 | 7/10 |
| Libero Sim | Close Top Drawer of Cabinet | 10 | 10/10 |
| | Turn on Stove | 10 | 9/10 |
| | Pick and Place Soup into Basket | 50 | 9/10 |

Table 3: User study comparing OPEN TEACH with baselines when used by experts and new users.

| Task | Success Rate | | | | Median completion time for successful demonstrations (in s) | | | |
| --- | --- | --- | --- | --- | --- | --- | --- | --- |
| | New User | | | Expert | New User | | | Expert |
| | **Holo-Dex** | **AnyTeleop** | **Open Teach** | **Open Teach** | **Holo-Dex** | **AnyTeleop** | **Open Teach** | **Open Teach** |
| Flip cube | 1 | 1 | 1 | 1 | 6.58 | 13.71 | 5.5 | 2.85 |
| Pinch Grasp | 0 | 0.2 | 0.8 | 1 | 17.49 | 18.94 | 18.72 | 3.71 |
| Pour | N/A | N/A | 0.4 | 0.8 | N/A | N/A | 40.97 | 14.83 |
| Pick and Place | N/A | N/A | 0.8 | 0.8 | N/A | N/A | 23.57 | 11.875 |
| Open box of mints | N/A | N/A | 0.5 | 1 | N/A | N/A | 32.21 | 20.45 |

## 4.2 Imitation Learning with OPEN TEACH Data

Here, we describe the algorithms used for learning policies on data collected through OPEN TEACH.

1. **Franka-Allegro:** We record both visual and tactile data for this setup. The policies are trained using TAVI [66], a demonstration-guided residual RL algorithm that collects a few expert demonstrations and learns a robot policy using both visual and tactile data.

2. **Allegro Sim:** We only record visual data for this setup and train policies using FISH [21].

3. **LIBERO Sim [23]:** We only record visual data for this setup. The policies are trained using transformer-based BC with a GMM head [67] and action chunking [2].

## 4.3 How versatile is OPEN TEACH across robotic setups?

The primary idea behind OPEN TEACH is that given any robotic setup, a user can purchase an affordable off-the-shelf VR headset (in this case, Quest 3) and plug the headset and robot setup into the proposed framework to start teleoperating the robot without any additional hardware setup cost. To investigate its versatility, we use OPEN TEACH to teleoperate four different real world robotic setups, each having a different combination of a robot arm and end effector type — Franka Allegro, Kinova Allegro, a Bimanual setup with 2 xArm7 robots, and Hello Stretch for mobile manipulation. We also exhibit the applicability of OPEN TEACH in simulation through evaluations on 2 simulated environment suites — Allegro Sim and LIBERO Sim [23]. The frequency of teleoperation for each of the setups has been provided in Table 1. Table 7 provides a set of tasks performed on Franka-Allegro, Allegro Sim, and LIBERO Sim. A more comprehensive list of tasks, including those on the Kinova-Allegro, Bimanual, and Stretch setup have been provided in Fig. 1 and Appendix 5.2.2.

## 4.4 How successful are policies trained with OPEN TEACH?

Table 7 provides the success rates of policies learned using imitation learning across both the real-world and simulated setups. We use TAVI [66] to learn visuotactile policies on Franka-Allegro, and FISH [21] to learn visual policies on Allegro Sim. Similar to prior work [66, 21], these policies were learned within 20 minutes and achieved an average success rate of 82%, validating the high quality of the collected observation data. Behavior cloning policies on LIBERO Sim achieve an average success rate of 93%, confirming the high quality of the collected action data. Overall, the

learned policies achieve an average success rate of 86% across all tasks and robot morphologies. This highlights the effectiveness of OPEN TEACH in collecting data for policy learning.

### 4.5 Can OPEN TEACH be used for performing complex, long-horizon tasks?

In this section, we emphasize the efficacy of OPEN TEACH in executing a diverse array of complex, long-horizon tasks across various robotic configurations. Appendix 5.2.2 includes real-world task rollouts for the Bimanual, Franka-Allegro, Kinova-Allegro and Stretch setups. OPEN TEACH allows the collection of demonstrations for intricate tasks, ranging from high-precision activities like USB insertion to delicate movements such as slicing a cucumber. On the multi-fingered hand setup, we demonstrate a broad spectrum of tasks, encompassing extended activities like *placing objects in a basket and lifting it* to contact-rich manipulation scenarios like *opening a laptop* and *sliding a tea sachet off the table*. A detailed compilation of tasks performed in both real-world and simulated setups, along with more task rollouts, have been included in Appendix 5.2. Videos showcasing these task rollouts can be found on our project website.

### 4.6 How intuitive is the system for new users?

We assess the user-friendliness of OPEN TEACH through a comprehensive user study involving 15 new users. The study is conducted on the Franka-Allegro setup, chosen for its capacity to evaluate the system's performance on both the robot hand and the robot arm. Each participant is allocated a 10-minute practice session to familiarize themselves with teleoperating the robot setup. Following this, they perform five trials for each of three distinct tasks using Holo-Dex [13], AnyTeleop [35], and OPEN TEACH. To mitigate potential biases, the order of tasks is randomized for each user.

In Table 3, we present a comparative analysis of success rates and median completion times for new users across Holo-Dex, AnyTeleop, and OPEN TEACH for the tasks of cube flipping and pinch grasping. Given the relatively small user sample size, we chose to analyze the median rather than mean completion times to mitigate potential biases from outliers. Since the Holo-Dex and AnyTeleop baselines lack open-source code for arm retargeting, we were unable to evaluate them on tasks involving arm movements. Thus, our comparison is limited to the cube flipping and pinch grasping that do not require arm manipulation. On these tasks, OPEN TEACH demonstrates a higher success rate along with significantly reduced median time to complete tasks compared to the other baselines.

Table 3 also includes a comparison of success rates and median completion times between an expert and new users utilizing OPEN TEACH for all tasks. On average, new users exhibit a success rate that is 76% of the expert's and take $2.25\times$ longer to complete a task. Details regarding individual user performances are provided in Appendix 5.3. Intriguingly, some new users, despite their unfamiliarity with the framework, achieve comparable or superior performance to the experts in certain tasks. This observation highlights two factors: (1) the inherent variation in abilities among individuals, and (2) while our system is intuitive for new users, prolonged training leads to substantial improvement in their performance, with the potential for further enhancement with continued practice.

## 5 Limitations and Discussions

In this work, we introduce OPEN TEACH, an open-source unified framework designed to facilitate low-latency, high-frequency robot teleoperation. This versatile framework is tailored to accommodate diverse tasks and is compatible with a range of robot morphologies. However, we recognize a few limitations in this work: ($a$) OPEN TEACH relies on the accuracy of the in-built hand pose detection in the VR headset. Inaccuracies, particularly when fingers are occluded from view, can diminish the quality of hand tracking, posing challenges to teleoperation. ($b$) In specific instances, the pose detector on the Oculus board may misconstrue finger positions, leading to difficulties in executing gestures like gripper closing, which relies on precise pinches between fingers. Addressing these challenges through future research on hand pose detection and tracking holds the potential to enhance the ease and intuitiveness of teleoperation using VR headsets.

## Acknowledgments

We thank Mahi Shafiullah, Raunaq Bhirangi, Ben Evans, Yibin Wang, Venkatesh Pattabiraman, Haritheja Etukuru and Chenyu Wang for valuable feedback and discussions. This work was supported by grants from Honda, Meta, Amazon, and ONR awards N00014-21-1-2758 and N00014-22-1-2773. Lerrel Pinto is supported by the Packard Fellowship.

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

# Appendix

## 5.1 Framework details

### 5.1.1 Structure of the framework

We use ZeroMQ for networking between nodes. The OPEN TEACH framework is divided into two parts - *teleoperation* and *data collection*.

Teleoperation The teleoperator is divided into 5 components - Detector, Keypoint Transformer, Operator, Controller, and Visualizer. A brief description of each has been provided below.

1. **Detector:** Receives the hand keypoints from the Meta Quest 3 and publishes them to ZMQ sockets.

2. **Keypoint Transformer:** Subscribes the keypoints published by the detector and maps them to the robot pose.

3. **Operator:** Receives the robot pose from the keypoint transformer and the current robot state from the controller. The operator computes the robot's actions which are published to a ZMQ socket.

4. **Controller:** Subscribes an action from the operator and takes an action in the real or simulated environment. After taking the action, the controller publishes the current states of the environment for use by the operator.

5. **Visualizer:** Subscribes the RGB images from the camera process (or the environment in case of simulations) and puts it on the screen inside the VR app for visualization during teleoperation.

Data Collection A data recorder process subscribes sensor information (RGB and Depth images, tactile readings, timestamps) and robot-specific information (joint states, gripper states, timestamps) from the corresponding sockets and logs them in corresponding files. The data is then compiled together by matching the timestamps between the sensor information and robot-specific data.

### 5.1.2 Thumb Retargeting for Robot Hand

Section 3.3 provides details about the design of the OPEN TEACH wrapper for the robot hand. To recap, given the individual joint angles in the teacher's hand from the VR headset, the joint angles for the robot hand can be computed by directly commanding the robot's joints to the corresponding angles. This works well for all fingers except the thumb. Holo-Dex[13] deals with this by mapping the spatial coordinate of the teacher's thumb tip to that of the robot hand. Then an inverse kinematics solver is used to compute the joint angles of the thumb. In this case, the retargeting of the thumb is done in 2D space. These bounds, depicted in Fig. 4(a), define the thumb's reach limits. During retargeting, the thumb tip's zone on the 2D palm plane is detected, and a perspective transform from the human hand to the robot hand is applied, aligning the human thumb tip with the robot thumb tip on the 2D plane. However, using three separate bounds introduces jitters when the thumb tip transitions between zones and results in stagnancy when outside the bounds. Further, in Holo-Dex, the height of the robot thumb tip is fixed, allowing it to only move along the 2D space.

To address these challenges, OPEN TEACH employs a single, large zone spanning the entire thumb's workspace in 2D space(refer to Fig. 4(b)). When the thumb is within bounds, a perspective transformation retargets the human thumb tip to the robot thumb tip. In cases where the thumb goes out of bounds, the closest point within the bound is estimated and used for retargeting, avoiding stagnation. Additionally, instead of a fixed height, the thumb is allowed to move perpendicular to the 2D surface along the palm, mapping the height of the human thumb tip to the robot thumb tip based on maximum and minimum height bounds. This approach ensures smoother thumb motion and enables the performance of more complex tasks compared to Holo-Dex [13].

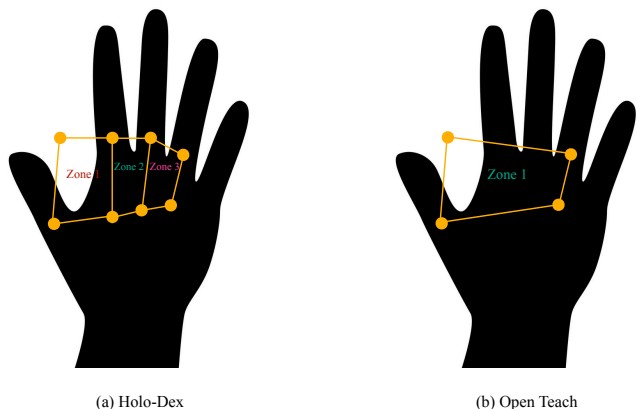

(a) Holo-Dex                  (b) Open Teach

Figure 4: Thumb retargeting difference

Table 4: Time

| Robot Setup | Task | Average time to collect a demo (in s) |
|---|---|---|
| Franka-Allegro | Open box | 45 |
| | Grasp sponge | 60 |
| | Pick up tea satchet | 60 |
| | Grasp object and twist | 35 |
| Kinova-Allegro | Unfold towel | 40 |
| | Open a pack of cream | 10 |
| | Open ketchup bottle | 40 |
| Bimanual | Uncap marker | 60 |
| | Sweep table | 60 |
| | Pour sprinkles in a bowl | 40 |
| Allegro Sim | Flip cube | 3 |
| | Flip sponge | 20 |
| | Pinch Grasp | 15 |
| LIBERO Sim | Close top drawer of cabinet | 10 |
| | Turn on stove | 25 |
| | Pick up and put soup can in the basket | 30 |

## 5.2 Task Details

### 5.2.1 Demo Collections times

Table 4 provides the average times required to collect a demonstration for 16 tasks across 3 real-world setups (Franka-Allegro, Kinova-Allegro, Bimanual) and 2 simulated environments(Allegro sim, LIBERO sim).

### 5.2.2 Task Descriptions

Fig. 5, Fig. 6, Fig. 7, Fig. 8, Fig. 9, and Fig. 10 provide rollouts of all the tasks performed both in the real world and in simulated environments. Each task rollout is labeled with the name of the task and a task description.

### 5.3 User Study

Following up from Section 4.6, we provide the success rate and average completion times for all 15 users for each task performed in Table 5 and Table 6 respectively. Each user roughly performed 3 tasks on average, with 5 trials for each task. As mentioned in Section 4.6, since the Holo-Dex [13]

and AnyTeleop [35] baselines lack open-source code for arm retargeting, we were unable to evaluate them on tasks involving arm movements. We observe a wide range of differences in success rates and average completion times demonstrating the inherent variations across users.

Table 5: Success rates for the user study conducted across 15 individuals. Each user roughly performs 3 tasks on average.

| User | Method | Success Rate (in 5 trials) | | | | |
|---|---|---|---|---|---|---|
| | | Flip Cube | Pinch Grasp | Pour | Pick and Place | Open Box of Mints |
| User 1 | Holo-Dex | 1 | 0 | - | - | - |
| | AnyTeleop | 0.8 | 0.2 | - | - | - |
| | Open Teach | 1 | 0.8 | 0.2 | - | - |
| User 2 | Holo-Dex | - | 0.2 | - | - | - |
| | AnyTeleop | - | 0.2 | - | - | - |
| | Open Teach | - | 0.8 | - | 0.8 | 0.8 |
| User 3 | Holo-Dex | 1 | 0 | - | - | - |
| | AnyTeleop | 1 | 0.2 | - | - | - |
| | Open Teach | 1 | 0.8 | - | - | 0.2 |
| User 4 | Holo-Dex | 1 | 0 | - | - | - |
| | AnyTeleop | 1 | 0.2 | - | - | - |
| | Open Teach | 1 | 0.8 | - | 0.6 | 0.4 |
| User 5 | Holo-Dex | - | 0 | - | - | - |
| | AnyTeleop | - | 0.6 | - | - | - |
| | Open Teach | - | 0.2 | 0.4 | 1 | - |
| User 6 | Holo-Dex | - | 0 | - | - | - |
| | AnyTeleop | - | 0.6 | - | - | - |
| | Open Teach | - | 0.8 | - | 0.2 | - |
| User 7 | Holo-Dex | - | 0 | - | - | - |
| | AnyTeleop | - | 0 | - | - | - |
| | Open Teach | - | 0.6 | 0.8 | 0.8 | 0.4 |
| User 8 | Holo-Dex | 1 | - | - | - | - |
| | AnyTeleop | 1 | - | - | - | - |
| | Open Teach | 1 | - | - | - | - |
| User 9 | Holo-Dex | - | 0 | - | - | - |
| | AnyTeleop | - | 0.4 | - | - | - |
| | Open Teach | - | 0.8 | 0 | - | 0.6 |
| User 10 | Holo-Dex | - | 0 | - | - | - |
| | AnyTeleop | - | 0.2 | - | - | - |
| | Open Teach | - | 0.6 | 0.4 | 1 | 1 |
| User 11 | Holo-Dex | 1 | - | - | - | - |
| | AnyTeleop | 1 | - | - | - | - |
| | Open Teach | 1 | - | - | 0.8 | 0.4 |
| User 12 | Holo-Dex | 1 | - | - | - | - |
| | AnyTeleop | 1 | - | - | - | - |
| | Open Teach | 1 | - | - | - | - |
| User 13 | Holo-Dex | 1 | - | - | - | - |
| | AnyTeleop | 1 | - | - | - | - |
| | Open Teach | 1 | - | 0.6 | - | - |
| User 14 | Holo-Dex | - | 0 | - | - | - |
| | AnyTeleop | - | 0.4 | - | - | - |
| | Open Teach | - | 0.6 | - | - | 0.8 |
| User 15 | Holo-Dex | 1 | - | - | - | - |
| | AnyTeleop | 1 | - | - | - | - |
| | Open Teach | 1 | - | 0.4 | - | - |

Table 6: Average completion times for successful trials for the user study conducted across 15 individuals. Each user roughly performs 3 tasks on average. *NS* denotes cases where no successes were achieved.

| User | Method | Average completion time for successful demonstrations (in s) | | | | |
|---|---|---|---|---|---|---|
| | | Flip Cube | Pinch Grasp | Pour | Pick and Place | Open Box of Mints |
| User 1 | Holo-Dex | 4.6 | NS | - | - | - |
| | AnyTeleop | 20.2 | 22.5 | - | - | - |
| | Open Teach | 5.4 | 18.6 | 66 | - | - |
| User 2 | Holo-Dex | - | 17.5 | - | - | - |
| | AnyTeleop | - | 18.9 | - | - | - |
| | Open Teach | - | 20.6 | - | 29.7 | 12.2 |
| User 3 | Holo-Dex | 5.4 | NS | - | - | - |
| | AnyTeleop | 18.3 | 7.8 | - | - | - |
| | Open Teach | 5.1 | 12.6 | - | - | 11.3 |
| User 4 | Holo-Dex | 11 | NS | - | - | - |
| | AnyTeleop | 13.2 | 31.4 | - | - | - |
| | Open Teach | 6.2 | 7.5 | - | 16.9 | 48.4 |
| User 5 | Holo-Dex | - | NS | - | - | - |
| | AnyTeleop | - | 11.4 | - | - | - |
| | Open Teach | - | 10.9 | 41.6 | 12.4 | - |
| User 6 | Holo-Dex | - | NS | - | - | - |
| | AnyTeleop | - | 12.7 | - | - | - |
| | Open Teach | - | 10.5 | - | 23.57 | - |
| User 7 | Holo-Dex | - | NS | - | - | - |
| | AnyTeleop | - | NS | - | - | - |
| | Open Teach | - | 19.1 | 21.49 | 49 | 37.8 |
| User 8 | Holo-Dex | 6.5 | - | - | - | - |
| | AnyTeleop | 5.4 | - | - | - | - |
| | Open Teach | 4.7 | - | - | - | - |
| User 9 | Holo-Dex | - | NS | - | - | - |
| | AnyTeleop | - | 49.9 | - | - | - |
| | Open Teach | - | 65.3 | NS | - | 32.21 |
| User 10 | Holo-Dex | - | NS | - | - | - |
| | AnyTeleop | - | 48 | - | - | - |
| | Open Teach | - | 30.8 | 40.3 | 48.7 | 21.3 |
| User 11 | Holo-Dex | 6.7 | - | - | - | - |
| | AnyTeleop | 11.5 | - | - | - | - |
| | Open Teach | 5.6 | - | - | 21.8 | 15.7 |
| User 12 | Holo-Dex | 6.2 | - | - | - | - |
| | AnyTeleop | 11 | - | - | - | - |
| | Open Teach | 3.8 | - | - | - | - |
| User 13 | Holo-Dex | 8.9 | - | - | - | - |
| | AnyTeleop | 14.2 | - | - | - | - |
| | Open Teach | 5.8 | - | 18.1 | - | - |
| User 14 | Holo-Dex | - | NS | - | - | - |
| | AnyTeleop | - | 49.9 | - | - | - |
| | Open Teach | - | 65.3 | - | - | 132.5 |
| User 15 | Holo-Dex | 13.2 | - | - | - | - |
| | AnyTeleop | 14.6 | - | - | - | - |
| | Open Teach | 6.3 | - | 53.1 | - | - |

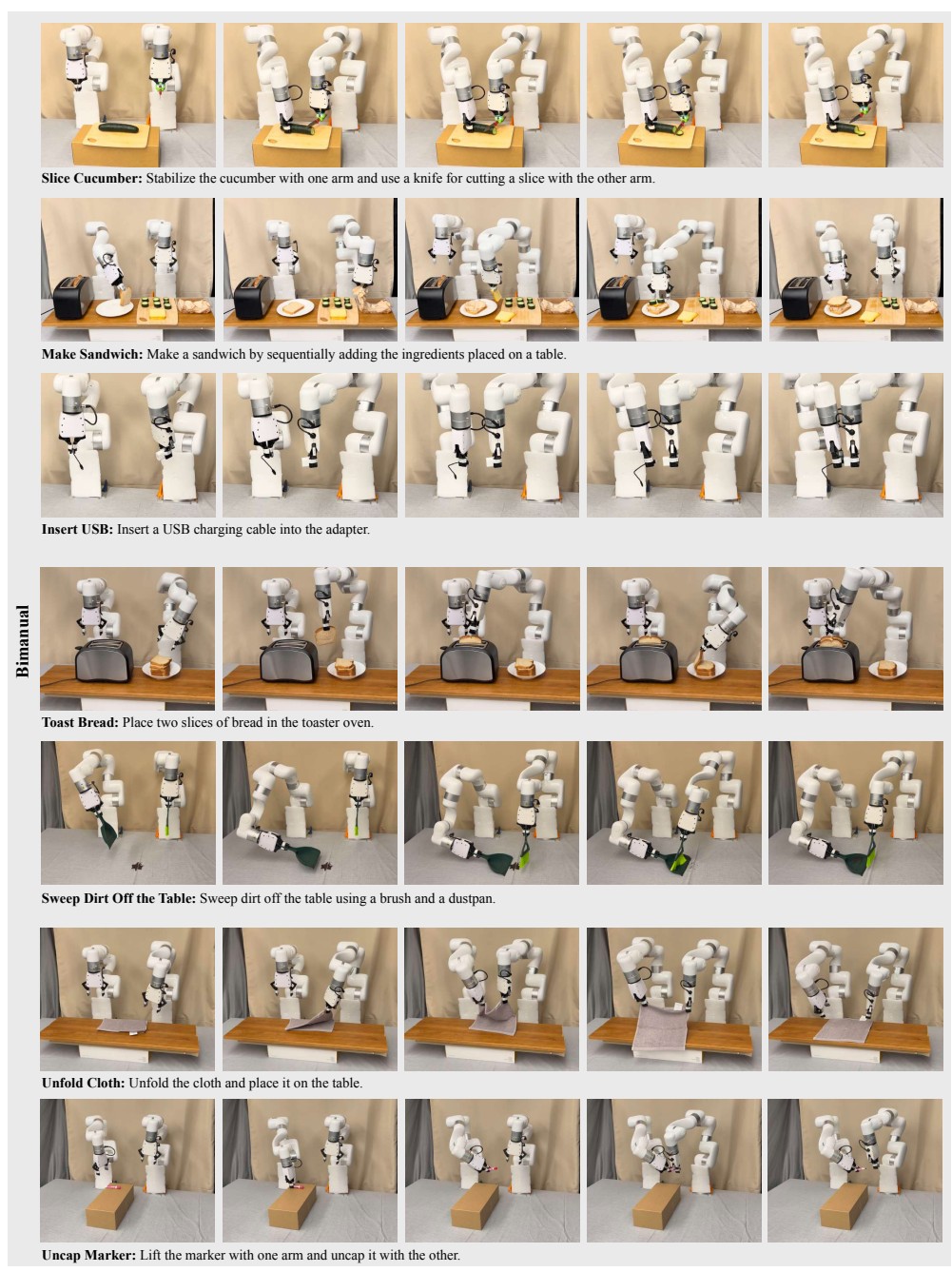

**Slice Cucumber:** Stabilize the cucumber with one arm and use a knife for cutting a slice with the other arm.

**Make Sandwich:** Make a sandwich by sequentially adding the ingredients placed on a table.

**Insert USB:** Insert a USB charging cable into the adapter.

**Toast Bread:** Place two slices of bread in the toaster oven.

**Sweep Dirt Off the Table:** Sweep dirt off the table using a brush and a dustpan.

**Unfold Cloth:** Unfold the cloth and place it on the table.

**Uncap Marker:** Lift the marker with one arm and uncap it with the other.

Figure 5: Real world task rollouts demonstrating the ability of OPEN TEACH to perform intricate, long-horizon tasks.

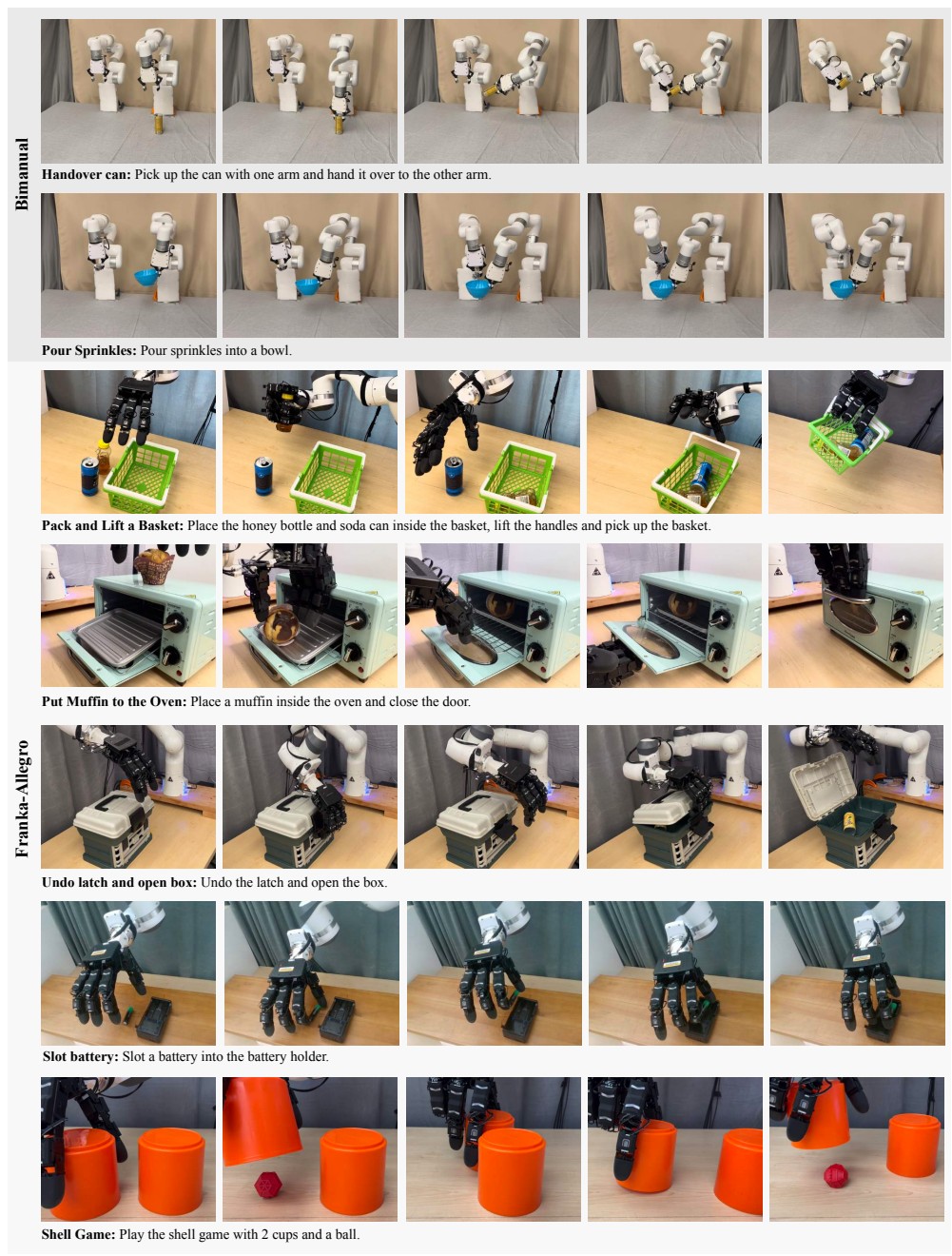

Figure 6: Real world task rollouts demonstrating the ability of OPEN TEACH to perform intricate, long-horizon tasks.

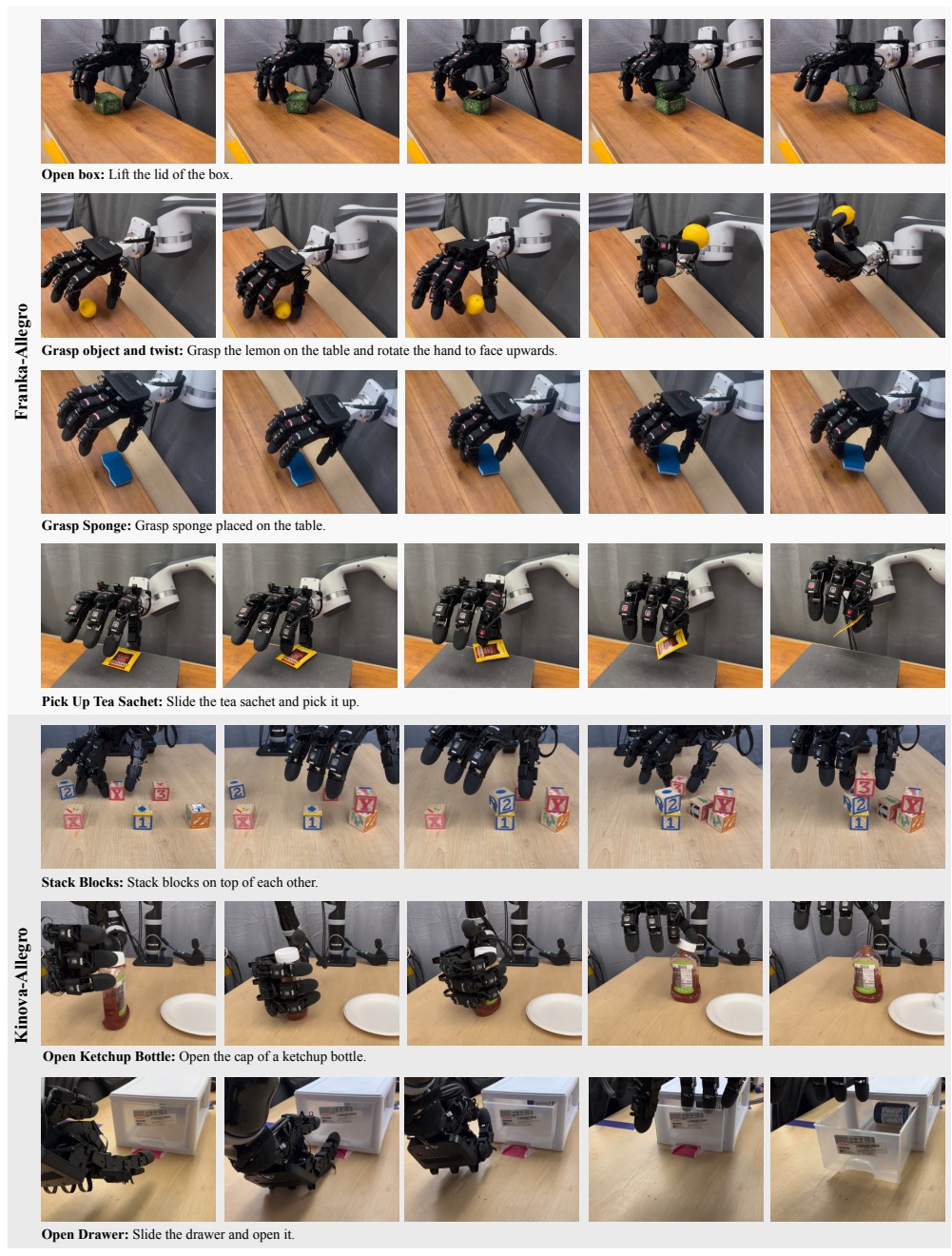

Figure 7: Real world task rollouts demonstrating the ability of OPEN TEACH to perform intricate, long-horizon tasks.

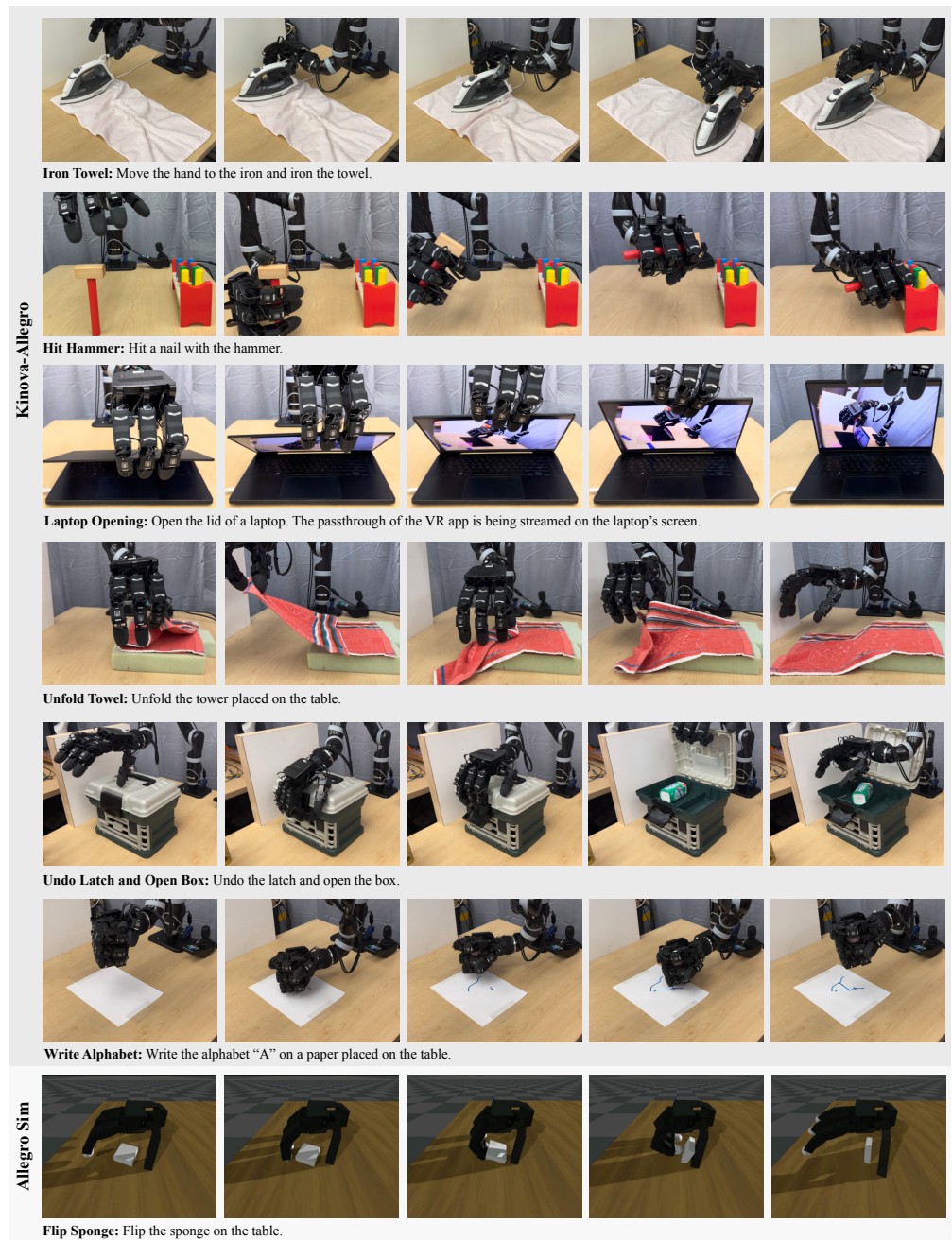

Figure 8: Real world task rollouts demonstrating the ability of OPEN TEACH to perform intricate, long-horizon tasks.

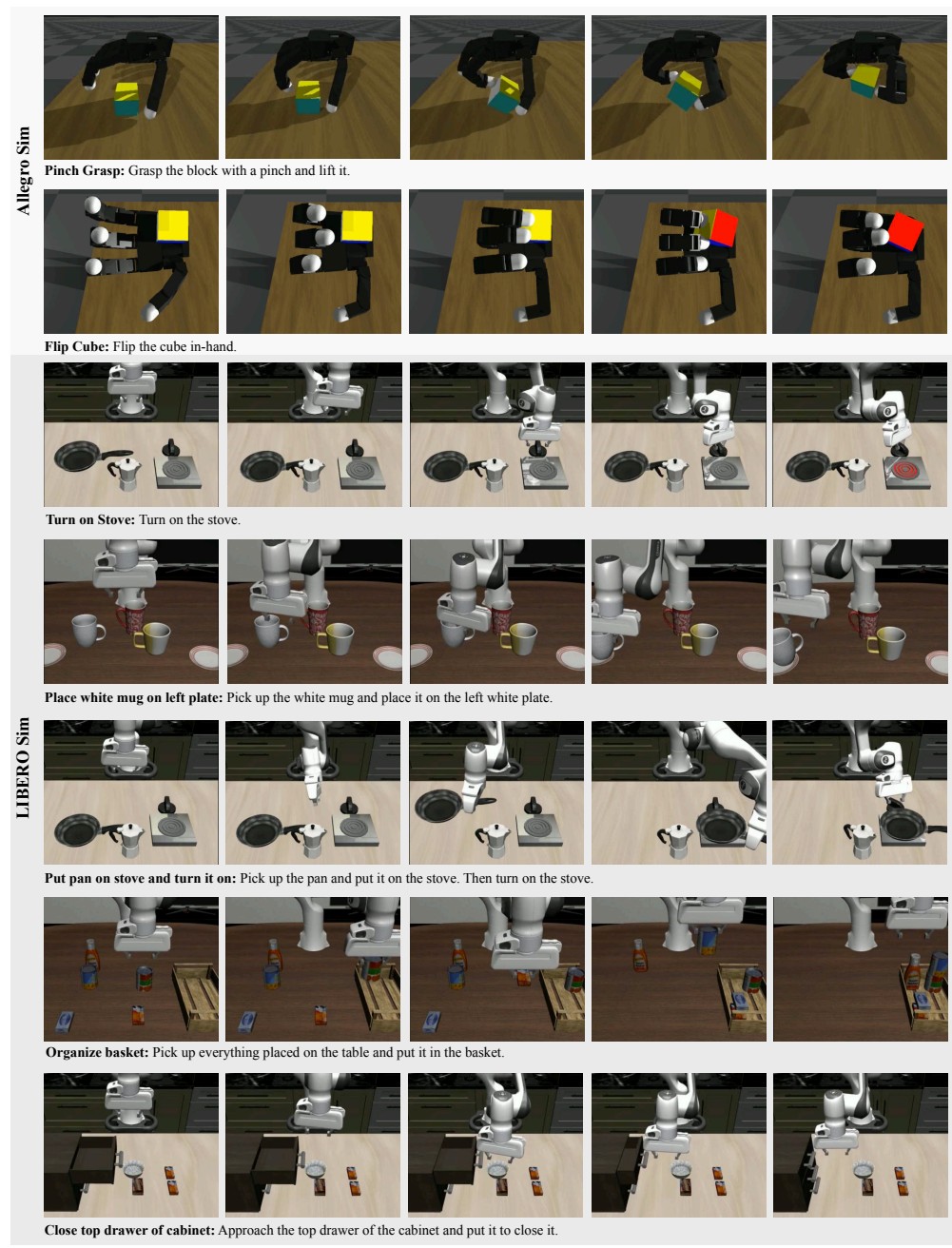

Figure 9: Real world task rollouts demonstrating the ability of OPEN TEACH to perform intricate, long-horizon tasks.

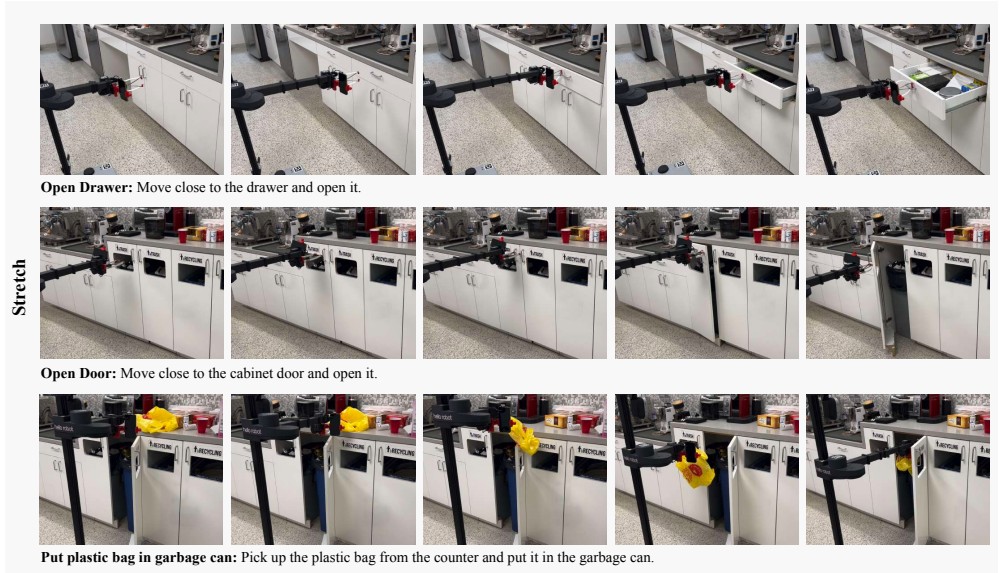

**Open Drawer:** Move close to the drawer and open it.

**Open Door:** Move close to the cabinet door and open it.

**Put plastic bag in garbage can:** Pick up the plastic bag from the counter and put it in the garbage can.

Figure 10: Real world task rollouts demonstrating the ability of OPEN TEACH to perform intricate, long-horizon tasks.

Table 7: Performance of policies learned on data collected through OPEN TEACH. FISH and BC were used to train policies for Allegro Sim and Libero Sim respectively. We report the mean and standard deviation for 25 evaluation trials across 3 seeds for each task.

| Robot Setup | Task | Number of Demos | Success Rate (25 trials) |
|---|---|---|---|
| Allegro Sim | Flip Cube | 6 | $0.97 \pm 0.03$ |
| | Flip Sponge | 6 | $0.79 \pm 0.05$ |
| | Pinch Grasp | 6 | $0.75 \pm 0.07$ |
| Libero Sim | Close Top Drawer of Cabinet | 10 | $0.96 \pm 0.03$ |
| | Turn on Stove | 10 | $0.95 \pm 0.04$ |
| | Pick and Place Soup into Basket | 50 | $0.77 \pm 0.02$ |



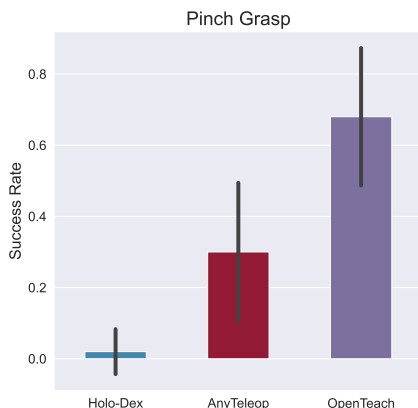

Figure 11: Success rates for the user study conducted across 15 individuals on 2 tasks - Flip Cube and Pinch Grasp. We report the mean and standard deviation for 3 methods - Holo-Dex, AnyTeleop, and Open Teach (Ours).

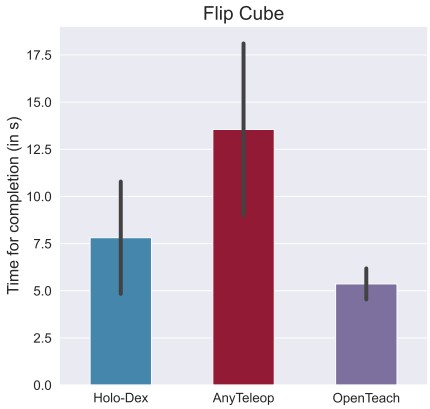
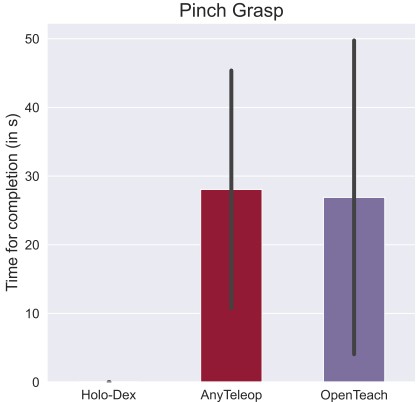

Figure 12: Average completion times for successful trials for the user study conducted across 15 individuals for 2 tasks - Flip Cube and Pinch Grasp. We report the mean and standard deviation for 3 methods - Holo-Dex, AnyTeleop, and Open Teach (Ours).

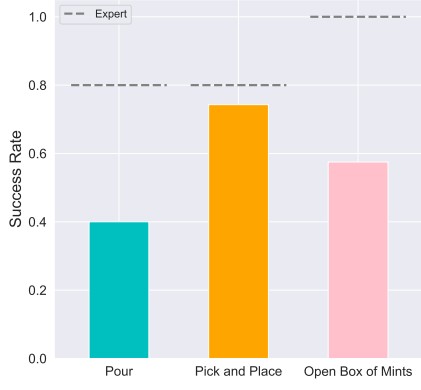

(a) Success Rate

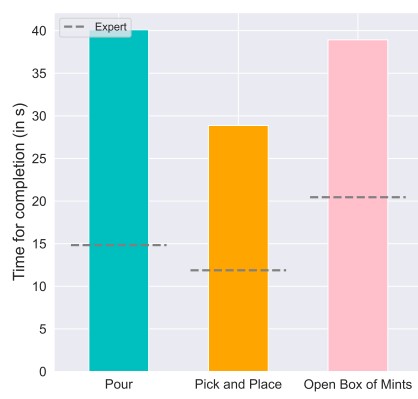

(b) Time for completion (in seconds)

Figure 13: We compare the (a) success rate, and (b) average completion time (in seconds) for using OPEN TEACH between an expert and 15 individuals participating in a user study. We report this comparison for 3 tasks - Pour, Pick and Place, and Open Box of Mints.

