# OpenReview forum: "OPEN TEACH: A Versatile Teleoperation System for Robotic Manipulation"
_robot-learning.org/CoRL/2024/Conference — CoRL 2024_

### Official Review · Reviewer_BA4K · 2024-07-21
**Trendy work but 'what' makes open teach better than other vision-based telepresence systems?**

**Originality:** 1
**Technical Quality:** 2
**Clarity Of Presentation:** 5
**Potential Impact:** 3
**Recommendation:** 3
**Confidence:** 4

**Review:**

Strengths:

- the paper is well written; very likely, the community will appreciate reading this paper since (a) design choices are well motivated, (b) key selling points are consistently outline throughout the paper, and (c) figures and website are helpful in demonstrating the advantage of the proposed system.

- experiments are well designed to answer various questions. I appreciated the extensive user study and comparison to AnyTeleop, which clearly shows advances over the state-of-the-art. Versatility, ability to perform imitation learning and long-horizon tasks are well shown, and extent of the experiments make the system-level contribution strong.

Weakness:

- (major) the impact of this paper would depend on how good the open-source frameworks are. Unfortunately, the open source codes and frameworks are not available for review.

This is a critical point since the whole purpose of the paper is to democratize vision-based teleoperation system, while being better than existing ones like AnyTeleop. I would also mention that there is no technical novelty associated, and its value lies in how open teach can serve the community as an infrastructure for robot learning and manipulation. For this reason, the open source code should have been included for a review, in order to judge its potential impact and also contribution to the robot learning field.

- (major) it is unclear what makes open teach better than other vision-based telepresence systems

The paper focuses on the major selling points. On the other hand, it does not disseminate "what" enables those selling points. In other words, what is the intellectual challenge faced, and further, what is the actual contribution of this paper?

**Quality Of The Limitations Section:**

3

**Questions For Rebuttal:**

If I think of vision-based telepresence systems, I always think of buggy hand gesture recognition. I agree that it is cheaper to build such systems, but my default choice would be something like ALOHA or more traditional DLR teleoperation systems like HUG or many of systems shown on recent AVATAR challenge. They are expensive, but more useful - with hardware, one can have all the advantages of open teach in supporting hands, manipulators, mobile manipulation and of course, real and simulation environments, and we do not have to suffer from vision-based softwares not being robust.

How accurate is the current hand gesture recognition in terms of usability? It was not satisfying that the paper mentions only at the end about such limitations. I think the tribute has to be paid to more traditional teleoperation systems throughout the paper other than ALOHA or recent learning-related papers, and comparison should be discussed.

**Robotics Focus:**

4

**Summary Of Paper:**

This paper presents a system called open teach, which is a vision-based teleoperation system for robotic manipulation. The paper claims that when compared to existing systems, within a single framework, open teach supports hands and manipulators, calibration-free and works across simulation and real world settings. Several manipulation tasks are demonstrated with the system. The user study confirms certain advancements over AnyTeleop, and the entire set up costs less than 500 dollars.

**Summary Of Recommendation:**

For me, the weakness outweighed the pros, thus recommending for a weak rejection. The paper is aiming to support the community through an open-source infrastructure. Thus, the open-source software and hardware should have been included for a review. Moreover, the paper should outline what is the main system level innovation from first principles, rather than focusing only on end-results. If these two points are addressed during the rebuttal, I am willing to change my recommendation since strenghts of the paper are also clear and have potential to serve the community.

---

### Official Review · Reviewer_rYe8 · 2024-07-22
**Review before Rebuttal**

**Originality:** 4
**Technical Quality:** 5
**Clarity Of Presentation:** 4
**Potential Impact:** 4
**Recommendation:** 3
**Confidence:** 5

**Review:**

### Strengths

OPEN TEACH demonstrates significant innovation in robotic teleoperation. Its key strength lies in its versatility, supporting multiple robot types across simulated and real environments. The system's affordability and open-source nature promote accessibility and reproducibility in robotics research.

The technical implementation is impressive, utilizing VR technology for immersive control with high-frequency operation (90Hz). This allows for real-time error correction, a notable advancement over existing systems. The comprehensive evaluation across 38 diverse tasks and user studies with 15 participants provides robust evidence of the system's effectiveness.

Importantly, OPEN TEACH addresses a crucial need in robotics for affordable, versatile data collection tools. The demonstrated compatibility of collected data with policy learning suggests the potential for accelerating progress in robot learning and manipulation tasks.

Overall, the reviewer appreciates the effort the authors spent to build such an open-source system with various robot support.

### Weaknesses

In general, I think this paper is good to be accepted from a robotics system perspective. But as a CoRL submission, it is also important to focus on the learning side and answer several robot learning questions:
1. What is the characteristic of the data collected by OPEN TEACH, how can it be different from data collected via exoskeleton like ALOHA?
2.  How much OPEN TEACH data is necessary for training an imitation learning policy with moderate generalization capability?

**Quality Of The Limitations Section:**

3

**Questions For Rebuttal:**

1. The authors demonstrate OPEN TEACH's capability in data collection for robot learning. However, it would be valuable to discuss the unique characteristics of data collected via OPEN TEACH compared to data from exoskeleton-based systems like ALOHA. What are the potential advantages or limitations of OPEN TEACH-collected data in terms of quality, diversity, or other relevant factors?

2. While the paper shows that policies can be trained using OPEN TEACH data, it would be beneficial to provide more insight into the data efficiency of this approach. Specifically, how much data collected through OPEN TEACH is typically required to train an imitation learning policy that exhibits moderate generalization capability? A quantitative analysis or comparison with other data collection methods would strengthen this aspect of the work.

**Robotics Focus:**

4

**Summary Of Paper:**

This paper introduce an affordable, open-source teleoperation system for robotic manipulation using a VR headset. It supports multiple robot types, offers high-frequency visual feedback, and works in both simulated and real environments. The system demonstrates versatility across various tasks, shows improved performance over existing frameworks, and produces data suitable for policy learning. OPEN TEACH aims to provide a user-friendly, comprehensive solution for robot teleoperation and data collection, addressing limitations of current systems and promoting broader adoption in robotics research.

**Summary Of Recommendation:**

The authors present a great system for robot teleoperation, the system contribution is enough to be accept. To further improve this paper, the author can consider explore the teleoperation system for downstream applications, e.g. how can this system impact imitation learning.

---

### Official Review · Reviewer_m2vB · 2024-07-22
**Good systems paper on teleop based on quest3. Works for several robots. Open source.**

**Originality:** 3
**Technical Quality:** 4
**Clarity Of Presentation:** 4
**Potential Impact:** 3
**Recommendation:** 2
**Confidence:** 5

**Review:**

**Strengths:**

1-Teleoperation is a relevant topic as teleop is being widely used for data collection for robot learning.

2-The system looks straightforward and easy to use, probably a good tool for researchers to use.

3-The paper is clear and easy to understand. Videos are also illustrative.


**Weaknesses:**

1- Other than using a new device as the quest 3 and being open source, the paper doesn’t bring points of novelty to research in teleoperation. The system is relatively straightforward, using the tracking from the quest 3, and doing a basic retargetting.


**Questions/Comments:**

1- It’s uncommon to publish table of results per each individual participant in the user study, as shown in the appendix. Rather, it’s a good idea to report all results with mean and variance, and compare conditions using the appropriate statistics. If you find value in reporting each data point, you can also present plots with all the points, mean and variance. Table 3 is a good example of results that need this improvement.

2-In general, plots would be helpful to visualize the results on the tables.

3-The paper mentions the systerm run at 90FPS but no details on running times are provided. What are the timings of processing the tracking, computing robot joint angles and sending commands?

4- Do operators experience physical fatigue? While the paper claims that the system is good for long tasks, I wonder how long the taks are and what the user experience like.

5-Table 2 presents results on only 10 trials. This is very low number of trials. Ideally you have more runs and can have a mean and a variance.

**Quality Of The Limitations Section:**

2

**Questions For Rebuttal:**

Questions above

**Robotics Focus:**

4

**Summary Of Paper:**

This paper presents the development of a teleoperation system that leverages the hand/fingers tracking available on the Quest 3 VR headset. The teleop system is designed to work with different robots and end effectors, single arm or bimanual tasks, and the library is open source.

**Summary Of Recommendation:**

The paper present a system for teleoperation that is adaptable to several robots and open source. As such, it would be useful to the community to have a report on the implementation of the system. However, from the research point of view for inclusion on the main program, it’s perhaps still a straightforward teleoperation system without significant new lessons. The paper is clear and easy to follow.

---

### Author Rebuttal · Authors · 2024-08-10

As per the instructions, we have added the global rebuttal as an "Official Comment" and are uploading the PDF associated with the global rebuttal here.

---

### Decision · Program_Chairs · 2024-09-04

**Decision:**

Accept

**Comment:**

The reviewers found some strengths in this submission, but also clearly articulated some questions for the rebuttal phase.
Thank you for your detailed responses.